# OpenReview forum: "BIOLOGICAL SYMMETRY CONSISTENCY: A FRAMEWORK FOR EVALUATING REPRESENTATIONAL MEANINGFULNESS IN COMPUTATIONAL BIOLOGY"
_ICLR.cc/2026/Workshop/LMRL — Submitted to ICLR 2026 Workshop LMRL_

### Official Review · Reviewer_Jd8n · 2026-02-10
**Review of Biological Symmetry Consistency (BSC)**

**Rating:** 4
**Confidence:** 4

**Review:**

This paper proposes BSC, a model-agnostic evaluation framework for representation meaningfulness in biology. It introduces a Symmetry Consistency Score (SCS) that compares embedding distances between original samples and symmetry-preserving transformed samples.

Strengths:
- Relevant problem
- Simple, intuitive evaluation metric
- Evaluation is independent of task-specific performance or model

Weaknesses:
- The claim of single-cell transcriptomic embeddings being invariant to gene index permutations is not justified as written for feature-based learning models. Permuting input gene order in, say, an MLP, without permuting the corresponding weight matrix will impact the representation. This holds true for the AE and VAEs used in transcriptomic evaluation. It is not clear whether the authors took this into account. While it is true that gene sets ("bag of genes") should be invariant to order, that does not make it true for models that take in genes as features.
- If the evaluated transformation does not align with the encoder’s inductive biases, SCS may primarily measure sensitivity to coordinate re-indexing rather than biological meaningfulness. The results in this paper show differences in SCS across models proving embeddings are not inherently invariant to order. Are these differences due to dataset-specific artifacts or the fact that the gene order was permuted without symmetrically permuting the model's weight matrix? Not specified in paper.
- This sentence does not adequately describe the evaluation protocol or answer the above questions: "Transformations were validated to ensure they do not alter known biological labels, with expert-rated transformation validation scores ranging 0–1."
- What is the overall framework score? Never clearly defined.
- Authors overstate generalizability given they only use 2 datasets and 4 fairly simple models for evaluation.
- No comparison of x and T(x) using methods such as CKA and SVCCA. Doing an evaluation of SCS vs. CKA/SVCCA would show if SCS is capturing something orthogonal/biologically useful instead of just being a proxy for existing methods. On the simpler side, one could use kNN overlap pre and post transform.

Conclusion: weak reject (4/10). Promising idea and solid motivation, but concerns regarding clarity, technical rigor, and superficial evaluations make me lean towards not accepting even as a "work-in-progress."

---

### Official Review · Reviewer_AZQw · 2026-02-16
**BSC: Valuable evaluation framework, with some points to clarify**

**Rating:** 2
**Confidence:** 5

**Review:**

### Summary

This paper introduces Biological Symmetry Consistency (BSC), a model-agnostic framework to evaluate biological representations by measuring the stability of embeddings under symmetry-preserving transformations. The Symmetry Consistency Score (SCS) normalizes the distance induced by transformations by the typical inter-sample distance, which makes comparisons possible across models (and even across modalities) without using task labels. Experiments on cellular morphology (BBBC021) and single-cell RNA (PBMC 3k), with 4 models, show that models with similar downstream accuracy can have quite different symmetry score, and that self-supervised and variational models tend to score higher.

### Strengths

1. Important motivation : the gap between downstream predictive performance and biological faithfulness is real and not so well addressed in practice. I like the idea of having a diagnostic metric that complements accuracy

2. Simple and practical : the SCS design is elegant and lightweight. The normalization by inter-sample distance is a good idea to make the score more comparable across models and datasets

3. Reasonable experimental practice : the statistical treatment is overall solid for a workshop paper (bootstrap confidence intervals, corrections for multiple tests....). The cross-modal rank correlation is also a signal that the approach is not completely dataset specific

4. Clear paper : the method and the motivation are explained clearly, and the figures help understanding

5. Good practical value : I can imagine using this framework quickly when I have several models with comparable downstream performance and I want a way to select one

### Weaknesses

1. Fabricated references = major concern
I attempted to verify all 19 references and found that at least 4 do not correspond to any real publication:

[6] P. Dwivedi, C. Agarwal, and H. Lakkaraju, “A framework for benchmarking the robustness of representation learning,” in International Conference on Learning Representations (ICLR), 2022.
[12] J. Ma and J. Z. Sun, “Evaluating single-cell data modality integration methods: a comparative analysis,” Genome Biology, vol. 22, no. 1, p. 341, 2021.
[16] K. von Chamier et al., “Self-supervised deep learning for rotation invariant cell profiling in microscopy images,” Nature Communications, vol. 14, p. 5328, 2023.
[19] Y. Zhu, Y. Wang, Z. Liu, and Y. Liang, “Bridging the gap between single-cell protein and transcriptome measurements across modalities and technologies,” Nature Biotechnology, vol. 41, pp. 1111–1124, 2023.

These fake ref might be LLM-generated citations: plausible titles, real author names recombined with non-existent papers, and credible info. This is a serious scholarly integrity concern. I strongly urge the authors to verify every single reference in their bibliography.

2. Citation format does not follow the ICLR template
The paper uses numbered bracket citations, whereas the ICLR template requires natbib author-year format with references listed alphabetically by author. Combined with the hallucinated references, this raises additional concerns about the level of care in the preparation of this manuscript.

3. Transcriptomics "symmetry" needs better framing : the paper presents gene index permutation as a biological symmetry. For me, this is not the same as rotation or reflection in imaging (where biology is truly invariant to orientation). Gene permutation is mainly a property of the data representation: genes have identities (names), so a model should not depend on arbitrary ordering. I think this part can still be useful, but it should be reframed more as an architectural invariance check rather than a biological symmetry in the strict sense

4. Limited experimental data : only 2 datasets and 4 rather simple models (CNN, simCLR, AE, VAE). For a tiny-paper this is acceptable as proof-of-concept, but it would be stronger with at least a few more baselines (PCA, random embeddings...) and ideally some foundation models mentioned in the introduction

5. Table 1 is unclear : the Baseline column is not properly defined. I did not understand what method produces these baseline values, and what exactly is measured (especially Overall Framework Score). I suggest either removing the table or clarifying with precise definitions and citations

6. Correlation claim is overstated : the paper reports ρ=0.31 (p=0.42) between SCS and accuracy and interprets it as "orthogonal properties". But this correlation is computed over only 4 models, so it is not really informative. I would strongly suggest to remove this claim or to state it as very preliminary

7. Theoretical section is a bit thin : Section 4 mostly restates nice properties of the metric, but does not bring much theoretical insight (e.g., no formal analysis of statistical behavior, no deeper connection to symmetry or group structure)

### Questions and suggestions

1. Could you clarify the distinction between biological symmetries and data format invariances (especially for transcriptomics)? In practice, what does SCS measure for models that take named gene inputs?

2. Did you test sensitivity to the choice of distance?

3. What happens when the inter-sample distance in the denominator is small (very homogeneous datasets)? Does SCS become unstable?

4. Can you add a couple of calibration baselines (random embeddings, PCA) to interpret the absolute magnitude of SCS?

5. For future modalities (proteomics, spatial transcriptomics), can you propose more concrete examples of transformations that are truly symmetry-preserving?

6. Can you confirm that all references have been individually verified? Please provide DOIs or URLs for references [6], [12], [16], and [19].

### Minor issues
- the n=4 correlation analysis should be qualified (or removed)
- some experimental details are missing (how rotations are sampled, exact permutation procedure)
- a short statement about code availability and reproducibility would help

### Overall assessment
I think this is a good workshop contribution. The main idea (SCS) is useful, easy to implement, and addresses a real pain point in biological representation learning. The imaging part is convincing, and the message "downstream perf is not enough" is important. The main weaknesses are mostly about framing (transcriptomics symmetry) and limited baselines, and I believe they are fixable with a clearer discussion and small additions.

However, the presence of at least 4 fake references out of 19 is a serious concern that undermines the credibility of this submission. Combined with the incorrect citation format and several sections that read as LLM-generated (Table 1 with undefined metrics, the thin theoretical section restating obvious properties), this raises questions about how much of the manuscript was produced by a language model without human verification. While ICLR does not prohibit the use of LLMs for writing assistance, the authors bear full responsibility for the accuracy and integrity of the content, including all references.
I would be willing to raise my score if the authors:
- verify and correct all references
- reformat citations to match the ICLR template

As it stands, the fabricated references alone raise serious concern.

---

### Official Review · Reviewer_96Xn · 2026-02-24
**Review of BIOLOGICAL SYMMETRY CONSISTENCY: A FRAMEWORK FOR EVALUATING REPRESENTATIONAL MEANINGFULNESS IN COMPUTATIONAL BIOLOGY**

**Rating:** 5
**Confidence:** 3

**Review:**

**1. Summary**

The paper proposes a new evaluation framework, Biological Symmetry Consistency (BSC), and a corresponding metric, the Symmetry Consistency Score (SCS). The goal is to evaluate learned representations based on their invariance to symmetry-preserving biological transformations (e.g., morphological rotations/reflections for cells, or gene permutations for transcriptomics), independent of downstream predictive accuracy.

**2. Strengths**
* **Important Motivation:** Evaluating embeddings for true biological faithfulness, rather than over-relying on downstream task performance, is an important and highly relevant goal for representation learning in computational biology.

**3. Major Weaknesses & Critiques**
* **Confounding of Training Augmentations vs. Inherent Representation:** While embeddings that are invariant to biologically preserving transformations are highly desirable, this property is usually explicitly built into the training regime via data augmentations, architectural decisions, or regularization terms in the loss function. For example, SimCLR (which is used to demonstrate the success of self-supervised learning here) learns invariant representations *by design* precisely because image rotations and reflections are part of its standard contrastive training regime.
* **Lack of Baseline Clarity:** Following the above point, it is not clear if the baseline models explored here (e.g., the supervised CNNs) were trained to have these invariances via identical data augmentation. If the self-supervised models were trained with rotational augmentations and the supervised models were not, the conclusion that self-supervised models inherently learn better biological symmetries is unsupported; the experiment simply tests whether the models remembered their own training objectives.
* **Unexplained Metrics (Table 1):** It is unclear how the values in Table 1 were arrived at. For example, what is the mathematical formulation for the "Overall Framework Score"? How were the baseline values arrived at? The paper currently lacks the methodological definitions necessary to justify these reported numbers, making it difficult to assess the validity of the results.
* **Limited Scope:** To truly prove the utility of this framework, the authors should include more baseline models (particularly modern foundation models) and define a wider array of transformations across more modalities.

**4. Minor Comments & Formatting**
* **Redundancy:** There are several instances of redundant text that should be cleaned up. For example, the authors define BSC and SCS multiple times throughout the short text. Additionally, BSC/SCS is referred to as a "complementary diagnostic" several times. A thorough editing pass would improve readability.

**5. Questions for the Authors**
1. Were the baseline models (e.g., the supervised models) trained using the exact same data augmentations (e.g., rotations/reflections) as the self-supervised models like SimCLR? If not, how do you decouple the effect of explicit training augmentations from the inherent quality of the self-supervised representation?
2. Can you explicitly define the methodology, formulas, and baselines used to calculate the "Transformation Validation" and "Overall Framework Score" values presented in Table 1?

---

### Meta-Review · Area_Chair_FFLU · 2026-02-28

**Recommendation:** Reject
**Confidence:** 5

**Metareview:**

This looks like it could be usual work in the future but the fabricated references pointed out by reviewer AZQw are clear grounds for rejection.

---

### Decision · Program_Chairs · 2026-03-02

**Decision:**

Reject

**Comment:**

Please see the meta-review.